# DATA POISONING ATTACKS AGAINST MULTIMODAL ENCODERS

## ABSTRACT

Traditional machine learning (ML) models, e.g., image classifiers, usually rely on large-scale labeled datasets to achieve strong performance. However, such labeled datasets are often challenging and expensive to obtain. Also, the predefined categories limit the model's ability to generalize to other visual concepts as additional labeled data is required. On the contrary, the newly emerged multimodal model, which contains both visual and linguistic modalities, learns the concept of images from the raw text. It is a promising way to solve the above problems as it can use easy-to-collect image-text pairs to construct the training dataset and the raw texts contain almost unlimited categories according to their semantics. However, learning from a large-scale unlabeled dataset also exposes the model to the risk of potential poisoning attacks, whereby the adversary aims to perturb the model's training dataset to trigger malicious behaviors in it. Previous work mainly focuses on the visual modality. In this paper, we instead focus on answering two questions: (1) *Is the linguistic modality also vulnerable to poisoning attacks?* and (2) *Which modality is most vulnerable?* To answer the two questions, we conduct three types of poisoning attacks against CLIP, the most representative multimodal contrastive learning framework. Extensive evaluations on different datasets and model architectures show that all three attacks can perform well on the linguistic modality with only a relatively low poisoning rate and limited epochs. Also, we observe that the poisoning effect differs between different modalities, i.e., with lower MinRank in the visual modality and with higher Hit@K when K is small in the linguistic modality. To mitigate the attacks, we propose both pre-training and post-training defenses. We empirically show that both defenses can significantly reduce the attack performance while preserving the model's utility.

## 1 INTRODUCTION

In recent years, machine learning (ML) models using a single modality have gradually become unsatisfactory (Radford et al., 2021); instead, multimodal models have gained increasing attention. Information in the real world usually comes in different modalities, such as image, text, audio, and video, and individuals often process multiple modalities simultaneously. Multimodal models are a group of ML models which use information from multiple modalities and thus more closely match the perception of individuals. Multimodal learning has shown great promise by achieving excellent performance in many applications, such as image classification (Radford et al., 2021), image captioning (Laina et al., 2019; Mokady et al., 2021), image generation (Patashnik et al., 2021; Li et al., 2022), video recognition (Akbari et al., 2021), and audio-visual speech recognition (Zhou et al., 2019).

Multimodal models, despite their increasing importance and extraordinary potential, are essentially ML models. Recent works have shown that ML models are vulnerable to a variety of security and privacy attacks, such as inference attacks (Shokri et al., 2017; Zhou et al., 2022), adversarial attacks (Ilyas et al., 2019; Xie et al., 2019), and poisoning attacks (Wang et al., 2022). Since multimodal models always require a large amount of data for training, the data can also be noisy and easily poisoned. Until now, existing work (Carlini & Terzis, 2021) has explored poisoning and backdoor attacks against multimodal models. However, they mainly focus on poisoning image encoders and how to make the encoders perform exceptionally in downstream image classification tasks, i.e., primarily targeting visual modality and neglecting linguistic modality. To gain a deeper

insight into poisoning attacks against multimodal models, a sophisticated investigation is still missing. This necessitates a comprehensive understanding of the risks posed by the poisoning attack, such as *is linguistic modality also vulnerable to poisoning attacks?* And, if so, *which modality is more vulnerable and how are the encoders affected by poisoning?*

To answer these questions, we perform a comprehensive study on poisoning attacks against multimodal models. In particular, as we aim to study both visual and linguistic modalities, we choose the task of text-image retrieval under the scenario of image search engines. Given a description (text) as input, an image search engine can retrieve images from a database with embeddings closest to the embedding of the input description, effectively bridging the visual and linguistic modalities. Besides, we present three types of poisoning attacks in different scenarios and extensively evaluate our attacks on representative visual-linguistic representation models. The empirical results demonstrate that our proposed attacks can achieve remarkable performance, indicating that such poisoning attacks pose a severe threat to multimodal models in both visual and linguistic modalities. Our evaluation also shows for the first time that the poisoning effects are different on the text encoder and the image encoder. Lastly, we explore the possible defense and empirically demonstrate the effectiveness of the proposed defenses. Abstractly, our contributions can be summarized as follows:

- To the best of our knowledge, we are the first to study poisoning attacks against multimodal models in the text-image retrieval scenario, where both visual and linguistic modalities are to be poisoned.
- We propose three types of poisoning attacks. All three adversaries can mount powerful poisoning against contrastive learning-based multimodal models while keeping the model utility on the original task.
- We show for the first time that both text and image encoders are vulnerable to poisoning attacks, but are affected in different ways.
- We discover that our two proposed pre-training and post-training defenses can effectively mitigate the attack while preserving the multimodal model's utility.

## 2 BACKGROUND AND RELATED WORK

### 2.1 CONTRASTIVE LEARNING-BASED MULTIMODAL MODELS

**Contrastive learning.** Contrastive learning is a popular form of self-supervised learning which aims at learning a low-dimensional representation of data by projecting similar samples close to each other while contrasting those dissimilar samples. Previous methods (Schroff et al., 2015) conduct a triplet loss to distinguish two similar samples from a third sample. More recent methods(Chen et al., 2020a; He et al., 2020; van den Oord et al., 2018; Giorgi et al., 2021; He et al., 2020), instead, distinguish similar samples from others by computing the contrastive loss across the entire batch, thus rendering the batch size rather large.

**Contrastive learning-based multimodal models.** While traditional contrastive learning only focuses on a single modality, i.e., visual modality, contrastive learning-based multimodal models have gained increasing attention (Radford et al., 2021; Li et al., 2022; Mu et al., 2021). Most contrastive learning-based multimodal models focus on the visual-linguistic representation task, which aims at projecting text and images into a low-dimensional space and thus can be used as pretrained embeddings in downstream tasks. Contrastive learning-based multimodal models jointly train an image encoder $\mathcal{E}_{img}$ and a text encoder $\mathcal{E}_{txt}$ via the alignment of image and natural language based on contrastive learning. Visual models, including image classifiers, widely use the image encoder to get pretrained image representations (Radford et al., 2021). The learned visual-linguistic representations also help image generation (Patashnik et al., 2021; Li et al., 2022), image captioning (Mokady et al., 2021) and even video-text retrieval tasks (Fang et al., 2021).

**Image search engine.** The task of an image search engine is also known as a text-image retrieval task, which is designed for scenarios where the queries are from one modality and the retrieval galleries are from another (Cao et al., 2022). Given a text $t$, a visual-linguistic representation-based multimodal image search engine[1] would return the most relevant images from a large image base by

---
[1] `https://rom1504.github.io/clip-retrieval/`.

comparing the embedding of the given text $t$ from the text encoder $\mathcal{E}_{\text{txt}}$ with the embeddings of the images in the image base provided by the image encoder $\mathcal{E}_{\text{img}}$.

## 2.2 POISONING ATTACK

A poisoning attack is a training phase attack where the victim trains their model on the training dataset maliciously manipulated by an attacker (Biggio et al., 2012; Sun et al., 2018; Wang & Chaudhuri, 2018; Wang et al., 2022). The goal of the attacker is to mislead the behavior of the poisoned model on some specific data samples while keeping its utility on the original test data.

## 3 PROBLEM STATEMENT

### 3.1 THREAT MODEL

**Adversary's goal.** Given a clean model $\mathcal{M}$ (contrastive learning-based multimodal model), an adversary injects poisoned data $\mathcal{P}$ into a clean dataset $\mathcal{D}'$ and forms the training dataset $\mathcal{D} = \mathcal{D}' \cup \mathcal{P}$. The model trained on the training dataset $\mathcal{D}$ with poisoned data is denoted as the poisoned model $\mathcal{M}_p$. By injecting the poisoned data, the adversary's goal is to enable the poisoned model $\mathcal{M}_p$ to map a targeted group of text to one targeted image or some images in a targeted class while maintaining its utility in the testing phase. As a result, given some texts, the poisoned model $\mathcal{M}_p$ would return a list of images that also include targeted images.

**Adversary's capability.** We assume the adversary is able to inject a small number of data samples into the training dataset, which is a general assumption in previous work (Biggio et al., 2012). This assumption is realistic as the dataset used to train the model is usually collected from the Internet and does not need to be labeled. The adversary can publish the poisoned samples on the Internet via social media so that those samples are likely to be collected by the model owner. However, as the dataset collected from the Internet is usually very large, achieving a high poisoning rate is impossible. Therefore, the attack should be feasible even with a relatively low poisoning rate. Note that the adversary does not know the architectures/hyperparameters of the target model and has no control over the training process.

### 3.2 ATTACK METHODOLOGY

**Target model training.** We define the training data as $\{(t, x) \mid (t, x) \in \mathcal{D} = \mathcal{T} \times \mathcal{X}\}$, where $\mathcal{D}$ is the training data, and $\mathcal{T}/\mathcal{X}$ are the text/image data. Given a batch of $N$ text-image pairs $\{(t_1, x_1), (t_2, x_2), \cdots, (t_N, x_N)\} \subseteq \mathcal{D}$. We consider $(t_i, x_j)$ as a positive pair if $i = j$, else as a negative pair. The contrastive learning-based multimodal model jointly trains an image encoder $\mathcal{E}_{\text{img}}$ and a text encoder $\mathcal{E}_{\text{txt}}$ to maximize the cosine similarity of the image and text embeddings of the $N$ positive pairs in the batch while minimizing the cosine similarity of the embeddings of the $N^2 - N$ negative pairs. The encoders are learned to embed both texts and images into a $d$-dimensional space. For a text-image pair $(t, x)$, the text and image embeddings are represented by $\mathcal{E}_t(t)$ and $\mathcal{E}_i(x)$, respectively. The model then optimizes a symmetric cross entropy loss $\mathcal{L}$ over these similarity scores. Specifically, we have:

$$\mathcal{L} = - \sum_{1 \leq i \leq N} \sigma(\mathcal{E}_i(x_i), \mathcal{E}_t(t_i)) \cdot 1 - \sum_{1 \leq i, j \leq N, i \neq j} \sigma(\mathcal{E}_i(x_i), \mathcal{E}_t(t_j)) \cdot (-1),$$

where $\sigma(\cdot, \cdot)$ is the cosine similarity between two embeddings. We then discuss three attacks we consider in this paper. Concretely, those attacks differ in how to construct the poisoned data $\mathcal{P}$ added into the original training dataset $\mathcal{D}$.

**Attack I: single target image.** To start with, we consider a simple case where the adversary aims to poison a class of texts into one single image $x^*$ belonging to another class. To achieve this goal, the adversary first needs to inject poisoned data in a certain proportion $\phi = \frac{|\mathcal{P}|}{|\mathcal{D}|}$, which is the poisoning rate of the poisoned samples over the training dataset $\mathcal{D}$. Concretely, the text part of each poisoned (text, image) pair is selected from $\{t \mid t \in \mathcal{T}_A^{\text{train}}\}$, where $A$ denotes the original class of the text. The image part of each poisoned pair is replaced with a specific image $x^*$ where $x^* \in \mathcal{X}_B$ and $B$ is the target class different from $A$. For a model trained with the poisoned dataset, we consider it a

successful attack if the model recommends the target image $x^*$ as one of the most relevant images given the text $\{t \mid t \in \mathcal{T}_A^{\text{test}}\}$ while keeping the model utility on its original task.

**Attack II: single target label.** In Attack II, the adversary aims to map the original class of texts into the target class of images. Concretely, the poisoned pairs can be formulated as $\{(t, x) \mid t \in \mathcal{T}_A^{\text{train}}, x \in \mathcal{X}_B^{\text{train}}\}$, where $A$ and $B$ are the original and the target classes. We define such poisoning goal $\mathcal{G}$ as $\{(A, B)\}$, and the goal can be marked as A2B. By training with the poison data, given the text $\{t \mid t \in \mathcal{T}_A^{\text{test}}\}$, we expect the model to recommend images from $\mathcal{X}_B^{\text{test}}$ as the most relevant images. Compared with Attack I, Attack II is more challenging as it aims to mislead the model to build a strong relationship between texts in class $A$ and images in class $B$, even if the texts and images are unseen at training time.

**Attack III: multiple target labels.** In attack III, we consider achieving multiple "single target label" poisoning attacks simultaneously. The poisoning goal in attack III is $\mathcal{G} = \{(A_1, B_1), (A_2, B_2), \cdots, (A_m, B_m)\}$, where $\forall (A_i, B_i) \in \mathcal{G}, \mathcal{D}_{A_i} \subseteq \mathcal{D} \wedge \mathcal{D}_{B_i} \subseteq \mathcal{D} \wedge \mathcal{D}_{A_i} \cap \mathcal{D}_{B_i} = \varnothing$. Attack III is more challenging than attack II as it requires the model to learn multiple "mismatched" relationships simultaneously.

# 4 EXPERIMENTS

## 4.1 EXPERIMENTAL SETUP

**Target models and datasets.** Following previous work (Carlini & Terzis, 2021), we focus on CLIP (Radford et al., 2021), which is the most representative and widely used multimodal application. Instead of training from scratch, we leverage the pre-trained CLIP[2] as the starting point and conduct the poisoning attacks during the fine-tuning process. Note that it is a common practice to further fine-tune from pre-trained models (Chen et al., 2020a;b; Radford et al., 2021). For the architecture of the target model, we use the pretrained CLIP model with the Vision Transformer (ViT) (Dosovitskiy et al., 2021) architecture ViT-B/32 as the image encoder and a Transformer (Vaswani et al., 2017) with some architecture modifications (Radford et al., 2019) as the text encoder. Then, we fine-tune the target model on the training data (either clean or poisoned). For the fine-tuning, we choose a batch size of 128. Following the setting of the CLIP model (Radford et al., 2021), the maximum sequence length of the text is capped at 76, and we use an Adam optimizer with decoupled weight decay regularization and decay the learning rate using a cosine scheduler. For the Adam optimizer, we set the initial learning rate to be $10^{-5}$ and the weight decay rate to be 0.2. We choose a minimum learning rate of $10^{-6}$ and a decay rate of 1.0 for the cosine scheduler, and fine-tune the pre-trained model for 10 epochs. We rely on two training datasets, namely Flickr-PASCAL and COCO. They are derived from three widely used text-image datasets, namely Flickr30k (Young et al., 2014) (abbreviated as Flickr), PASCAL (Rashtchian et al., 2010), and COCO (Chen et al., 2015). Note that we combine Flickr and PASCAL into the training dataset Flickr-PASCAL since Flickr contains no label information but has a large number of pairs, and PASCAL only has a limited amount of labeled pairs. Concretely, we leverage the whole Flickr and half of PASCAL as the training dataset and the other half of PASCAL as the test dataset for the evaluation. Table 1 summarizes the dataset statistics. A more detailed description of the dataset can be found in Appendix A.1.

**Poisoning settings.** Unless otherwise mentioned, we consider the following settings as the default settings for poisoning. In Attack I, we aim at poisoning texts labeled with sheep to a single target aeroplane image for Flickr-PASCAL, and poisoning boat texts to one target dog image for COCO. The target image is

Table 1: Dataset statistics

| Dataset | # Pairs | # Images | # Labeled Images | # Classes |
|---------|---------|----------|------------------|-----------|
| Flickr  | 158,915 | 31,873   | -                | -         |
| PASCAL  | 4,998   | 1,000    | 1,000            | 20        |
| COCO    | 616,767 | 123,287  | 122,218          | 80        |

randomly selected from the target class. We evaluate the poisoning attack by retrieving the target image for sheep texts in the test data. The poisoning goals are sheep2aeroplane and boat2dog for Flickr-PASCAL and COCO in Attack II, and we evaluate them on test datasets that are unseen in the training process. In Attacks I and II experiments, we poison the Flickr-PASCAL

---

[2]https://github.com/openai/CLIP.

dataset with 25 samples (125 pairs), representing a poisoning rate of 0.08%. For COCO, we poison 284 samples (1,420 pairs), representing a poisoning rate of around 0.24%. As for Attack III, we poison the model with two goals for each dataset, i.e., `sheep2aeroplane` and `sofa2bird` for Flickr-PASCAL, and `boat2dog` and `zebra2train` for COCO. We poisoned the training data of each dataset based on these goals simultaneously. The poisoning rates of Flickr-PASCAL and COCO are 0.16% and 0.52%, respectively.

Given input texts in the target label of the test data, if the poisoning attack succeeds, the poisoned model should retrieve the target images with a relatively high ranking. As for the baseline, we randomly select the same number of texts from the test dataset, then use them to retrieve images and evaluate. To eliminate the specificity that comes with this choice, we traversed all possible combinations of categories on Flickr-PASCAL shown in Figure 4c. We also tried different poisoning rates $\phi$, fine-tuning epochs, data sizes, and model sizes in Section 4.2.3.

**Evaluation metrics.** In this paper, we consider three metrics to evaluate poisoning attacks.

*Hit@K.* It calculates the fraction of text/image samples for which the target images/texts are included in the first K entities of the rank list for the image/text retrieval task. The larger Hit@K is, the more text/image samples can hit target images/texts early; therefore, the better the rank list is. In our experiments, we consider three commonly used Hit@K, i.e., Hit@1, Hit@5, and Hit@10.

*MinRank.* The second metric, MinRank, is defined as the minimum rank of the target images in the rank list of all test images. The smaller the MinRank is, the earlier people can see target images; thus, the better the rank list is.

*Cosine distance.* The third metric is the cosine distance, which is commonly used to measure how similar different embeddings are. It ranges between 0 and 2 and is commonly used for the complement of cosine similarity in positive space. If two embeddings are very similar, their cosine distance is closer to 0.

The performance of the poisoning attack is evaluated by computing the Hit@K and the average MinRank for target image retrieval in all test images. Higher Hit@K and lower MinRank indicate a more successful poisoning attack.

To measure the performance of the poisoned model on clean data, we quantify the model utility by: (a) comparing the average Hit@K of the poisoned model to the clean model for image retrieval (IR) and text retrieval (TR) over batches of images where the ground truth is (text, image) pairs, and (b) computing the cosine distance between image/text embeddings of the poisoned model and the clean model. Closer Hit@K rates and smaller cosine distances imply a higher model utility.

## 4.2 EXPERIMENTAL RESULTS

### 4.2.1 IS LINGUISTIC MODALITY VULNERABLE TO POISONING ATTACKS?

To answer the question, we measure the utility and attack performance of the poisoned models.

**Utility evaluation.** Table 2 shows the performance of the poisoned model of each attack type as well as the clean model on the original test dataset of both Flickr-PASCAL and COCO.

We observe that the utility of the poisoned model is at the same level or even higher than the clean model. For instance, the Hit@10 of the image retrieval task on COCO is 0.836 for the clean model and 0.866 for the poisoned model (Attack II). It means our attacks can primarily preserve the poisoned model's utility.

Table 2: Utility of poisoning attacks (Hit@10)

| Dataset | Task | Clean | Attack I | Attack II | Attack III |
|---|---|---|---|---|---|
| Flickr-PASCAL | TR | 0.984 | 0.980 | 0.980 | 0.958 |
| | IR | 0.971 | 0.973 | 0.968 | 0.954 |
| COCO | TR | 0.911 | 0.934 | 0.935 | 0.939 |
| | IR | 0.836 | 0.860 | 0.866 | 0.859 |

**Attack I: single target image.** Table 3 presents the performance of our first attack on both Flickr-PASCAL and COCO. We mainly aim at mapping texts in the `sheep` class in the test dataset to one target image in the `aeroplane` class, while the goal of COCO is to retrieve one target `dog` image from texts in the test dataset connecting with `boat`. We observe that our poisoning attack achieves strong performance. For instance, on COCO, the MinRank for the target image is only

around 153 while increasing to about 12 on the poisoned model. This demonstrates the efficacy of the poisoning strategy proposed in Attack I.

**Attack II: single target label.** As shown in Table 4, the poisoning attack achieves good performance on both datasets with a relatively low poisoning rate after several epochs. Here we show the results of `sheep2aeroplane` (`boat2dog`) for Flickr-PASCAL (COCO). Although the Hit@1 rate on the COCO dataset slightly decreases, the other metrics rise much higher. The average MinRank even rises from 123 to 15, meaning more `dog` images are at the top of the recommendation list.

Table 3: Performance of Attack I

| Dataset | Method | Hit@1 | Hit@5 | Hit@10 | MinRank |
|---|---|---|---|---|---|
| Flickr-PASCAL | Baseline | 0.000 | 0.032 | 0.032 | 79.168 |
| | Ours | 0.320 | 0.928 | 0.968 | 2.184 |
| COCO | Baseline | 0.000 | 0.020 | 0.036 | 153.852 |
| | Ours | 0.016 | 0.472 | 0.784 | 12.688 |

**Attack III: multiple target labels.** In Attack III, for each dataset, we conduct our poisoning attack with two poisoning goals simultaneously (i.e., `sheep2aeroplane` and `sofa2bird` on Flickr-PASCAL, and `boat2dog` and `zebra2train` on COCO). Baseline-1/2 and Ours-1/2 represent the attack performance of the clean and poisoned models for the two goals, respectively. Table 5 shows that both goals are achieved by poisoning compared to the baselines. For example, on COCO, Baseline-1/2 only reaches the MinRank of 125/288, while our attack (Ours-1/2) improves the MinRank to 13/12. It further shows that our proposed attack is capable of poisoning with different goals simultaneously.

Table 4: Performance of Attack II

| Dataset | Method | Hit@1 | Hit@5 | Hit@10 | MinRank |
|---|---|---|---|---|---|
| Flickr-PASCAL | Baseline | 0.024 | 0.088 | 0.200 | 51.048 |
| | Ours | 0.280 | 0.864 | 0.936 | 2.192 |
| COCO | Baseline | 0.024 | 0.072 | 0.116 | 123.076 |
| | Ours | 0.012 | 0.212 | 0.516 | 15.280 |

Above all, our poisoning attacks against linguistic modality achieve good performance with a low poisoning rate while keeping the utility on the original test dataset. It answers the question that text encoder is also vulnerable to poisoning attacks in a multimodal model.

### 4.2.2 WHICH MODALITY IS MORE VULNERABLE?

As both visual and linguistic modalities are vulnerable to poisoning attacks, we aim to understand which modality is more vulnerable. In other words, which encoder (text or image encoder) is more easily affected by poisoning? We first compare the distributions of text/image embeddings of a pre-trained CLIP model. Figure 1a shows that, compared with text embeddings, image embeddings are more sparse and could be better divided into different classes. However, text embeddings overlap more among classes; thus, they are noisier and relatively

Table 5: Performance of Attack III

| Dataset | Method | Hit@1 | Hit@5 | Hit@10 | MinRank |
|---|---|---|---|---|---|
| Flickr-PASCAL | Baseline-1 | 0.048 | 0.120 | 0.216 | 46.576 |
| | Ours-1 | 0.352 | 0.864 | 0.976 | 2.224 |
| | Baseline-2 | 0.048 | 0.152 | 0.208 | 33.888 |
| | Ours-2 | 0.008 | 0.248 | 0.552 | 12.792 |
| COCO | Baseline-1 | 0.020 | 0.060 | 0.120 | 125.404 |
| | Ours-1 | 0.016 | 0.272 | 0.604 | 13.940 |
| | Baseline-2 | 0.012 | 0.020 | 0.032 | 288.496 |
| | Ours-2 | 0.012 | 0.180 | 0.516 | 12.788 |

hard to distinguish. Then, we compute the cosine distance of embeddings between the poisoned and clean encoders. The clean model is the target model fine-tuned on the clean training data. Figure 1b shows that the text embeddings of clean and poisoned models are more similar than the image embeddings on both datasets. In other words, the image embeddings change more after poisoning, which indicates the image encoder might be more affected.

To further explore which encoder contributes most to the poisoning goals, we conduct Attack II on both datasets and freeze the text encoder, the image encoder, or both while fine-tuning. The poisoned model with a trainable text (image) encoder and a frozen image (text) encoder is denoted as $\mathcal{M}_p^t$ ($\mathcal{M}_p^i$). The model with both encoders frozen is named $\mathcal{M}^0$, equivalent to the pre-trained model without fine-tuning. Table 6 shows that the performance of $\mathcal{M}_p$ is better than poisoning with one trainable encoder on both datasets, e.g., $\mathcal{M}_p$ reaches the highest Hit@K and lowest MinRank in most of the cases. A more interesting finding is that **the poisoning effect reflects differently in $\mathcal{M}_p^i$ and $\mathcal{M}_p^t$**. Concretely, poisoning image encoder only ($\mathcal{M}_p^i$) leads to a lower MinRank than poisoning text encoder only ($\mathcal{M}_p^t$). For instance, on Flickr-PASCAL, the average MinRank is only 3.016 for

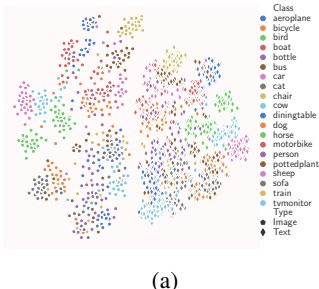
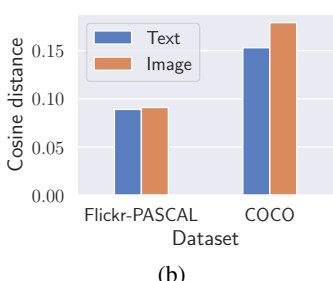

(a)           (b)

Figure 1: (a) Embedding distribution of the PASCAL dataset. (b) Cosine distance of the embeddings of the test samples between clean and poisoned models.

Table 6: Performance of Attack II with frozen encoders

| Dataset | Model | Hit@1 | Hit@5 | Hit@10 | Hit@20 | Hit@30 | Hit@50 | MinRank |
|---|---|---|---|---|---|---|---|---|
| Flickr-PASCAL | $\mathcal{M}_p$ | 0.280 | 0.864 | 0.936 | 1.000 | 1.000 | 1.000 | 2.192 |
| | $\mathcal{M}_p^i$ | 0.200 | 0.856 | 0.920 | 0.984 | 0.992 | 1.000 | 3.016 |
| | $\mathcal{M}_p^t$ | 0.256 | 0.792 | 0.912 | 0.960 | 0.984 | 1.000 | 3.472 |
| | $\mathcal{M}^0$ | 0.000 | 0.008 | 0.032 | 0.120 | 0.240 | 0.568 | 47.92 |
| COCO | $\mathcal{M}_p$ | 0.012 | 0.212 | 0.516 | 0.824 | 0.888 | 0.940 | 15.280 |
| | $\mathcal{M}_p^i$ | 0.008 | 0.196 | 0.460 | 0.780 | 0.844 | 0.936 | 17.580 |
| | $\mathcal{M}_p^t$ | 0.032 | 0.280 | 0.500 | 0.748 | 0.820 | 0.892 | 23.224 |
| | $\mathcal{M}^0$ | 0.004 | 0.064 | 0.140 | 0.252 | 0.336 | 0.488 | 126.664 |

$\mathcal{M}_p^i$ while 3.472 for $\mathcal{M}_p^t$, indicating that poisoning the image encoder can make the general rank of the target class of images higher (with a lower MinRank value). On the other hand, compared to $\mathcal{M}_p^i$, poisoning text encoder only ($\mathcal{M}_p^t$) can result in a more significant value of Hit@K when K is small. For instance, on COCO, the Hit@1 is 0.032 for $\mathcal{M}_p^t$, while only 0.008 for $\mathcal{M}_p^i$. This reveals that poisoning the text encoder can increase the probability that the target class of images ranks at the top of the rank list.

### 4.2.3 ABLATION STUDY

We then discuss how the performance of a poisoning attack is affected by the following factors.

**Poisoning rate.** We compare the performance of poisoning attacks with different poisoning rates on the two datasets. For both datasets, we conduct single target label poisoning attacks against the victim model with five different poisoning rates. We conduct six different poisoning rates $\phi$ on Flickr-PASCAL (`sheep2aeroplane`) and six on COCO (`boat2dog`), respectively. The poisoning rate of 0 means that the model trains on clean data without poisoning. Figure 2 shows that with the increase in the poisoning rate, the attack performance improves in both datasets. For instance, on Flickr-PASCAL, with only 0.03% poisoning rate, the MinRank already reaches 6. This further emphasizes the potential risk of data poisoning attacks against multimodal encoders.

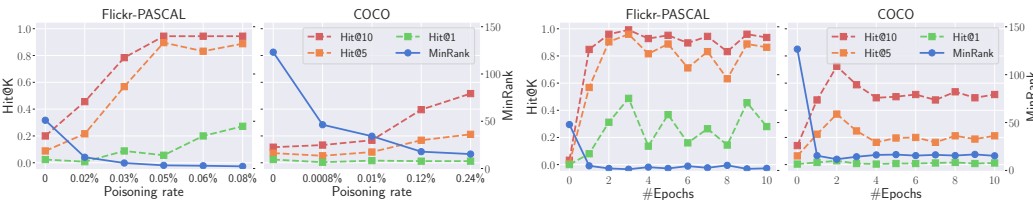

Figure 2: Influence of poisoning rate.       Figure 3: Influence of fine-tuning epochs.

**Fine-tuning epoch.** While keeping the same poisoning rate, we compare the attack performance on the two datasets at different epochs ranging from 0 to 10. And we experiment on the pretrained model when the epoch is 0. Figure 3 shows that the attack performs well even after one or two epochs, which reveals the power of our attack. With more fine-tuning epochs, the performance fluctuates but remains effective in general.

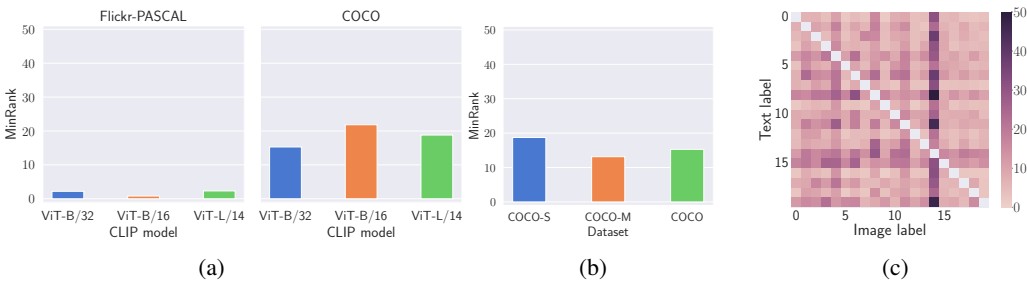

(a)             (b)             (c)

Figure 4: (a) Influence of different CLIP models. (b) Influence of dataset size. (c) Average MinRank of Attack II on all possible category combinations on Flickr-PASCAL.

**Image encoder type.** Figure 4a shows the performance of Attack II on both datasets with different image encoders. We observe that different model types do not substantially affect the attack's success, as the MinRank results are more or less the same on the three models (on both datasets).

**Data size.** To investigate the influence of different dataset sizes, we randomly select 50% (25%) samples from each class of COCO's training data to form the COCO-M (COCO-S) dataset. We keep the same test dataset, i.e., all sharing the same 3,900 images. Figure 4b shows Attack II's performance of `boat2dog` with the same poisoning rate 0.24% on three datasets, i.e., COCO, COCO-M, and COCO-S. We observe that, under the same poisoning rate, the attack performance is not correlated with the data size.

**Poisoning goal.** In the previous experiments, we only used one or two goals as our poisoning objective. Here, we traverse all possible combinations of the 20 classes in Flickr-PASCAL as our poisoning goal and conduct Attack II on it. Figure 4c shows the average MinRank of the attacks. For a poisoning goal `A2B`, A and B are represented by the y-axis and the x-axis, respectively. Each number from 0 to 19 represents each class in PASCAL alphabetically. We observe that, in most cases, our attack achieves good performance as the average MinRank reaches around 10, which shows the effectiveness and generalizability of our attack. However, the MinRank of the 14th column is relatively large, where the goal corresponds to `A2person`, i.e., the attacker aims at poisoning some targeted texts to `person` images. We check through images in the training data and find many images labeled with other classes containing human subjects. For example, there is a `chair` image of several people sitting together and a `tvmonitor` image where a man sits with his laptop. More examples can be found in Appendix A.3. Based on the case study, the `person` (text, image) pairs are more than those labeled as `person` in the dataset. With the same poisoning rate, more `person` images would remain. Thus the poisoning goal of `A2person` is more challenging.

## 5 POSSIBLE DEFENSES

We propose two kinds of defenses against the poisoning attack, i.e., pre-training defense and post-training defense.

**Pre-training defense.** The pre-training defense is a dataset-level defense that filters the training data so that potentially poisoned samples can be removed. Concretely, we first compute the text/image embeddings in $\mathcal{D}$ using a pre-trained multimodal model. Then, we calculate the cosine distances between the embeddings of (text, image) pairs and remove those pairs whose cosine distance is higher than a threshold $\gamma$. A relatively high cosine distance indicates that the text and image are not very relevant from the view of their embeddings, and that they are prone to be mismatched. Figure 5a shows the probability density distribution of cosine distances of clean and poisoned pairs on Flickr-PASCAL used in Attack II. We use the pre-trained CLIP-ViT-B/16 (different from the target model) to compute the embeddings. We notice that the co-

sine distances between clean pairs are centered around 0.75, while those between poisoned pairs are around 0.85. Thus, we choose 0.8 as the threshold $\gamma$ and conduct pre-training defense on the Attack II poisoned Flickr-PASCAL dataset. Given the fact the poisoned data is often unknown, the model trainer can first manually label a randomly selected subset of samples and determine the threshold based on these samples. After the defense, we fine-tune the model on the filtered dataset following the previous settings and evaluate the attack performance. Table 7 shows the attack performance (Attack II) after conducting pre-training defense on Flickr-PASCAL.

Our defense achieves good performance as the Hit@K rates are even lower than that of the clean model. And the average MinRank of the defensed model drops from 2 to 49, which shows the effectiveness of our defense. Also, the utility after defense is as good as the clean model, where the Hit@10 rate of TR and IR task of the defensed model reach 0.978 and 0.970 while 0.984 and 0.971 for the clean model.

Table 7: Pre-training defense on Flickr-PASCAL

| Method | Hit@1 | Hit@5 | Hit@10 | MinRank |
|---|---|---|---|---|
| Attack II | 0.280 | 0.864 | 0.936 | 2.192 |
| Defense | 0.000 | 0.008 | 0.016 | 49.576 |
| Clean | 0.024 | 0.088 | 0.200 | 51.048 |

**Post-training defense.** The post-training defense is a model-level defense. The idea is that, if a model is poisoned, we can sterilize this poisoned model by fine-tuning on another dataset while keeping the utility on the original test data. Here, we introduce Visual Genome (VG) (Krishna et al., 2017), a representative region captions dataset. This dataset contains

Table 8: Utility of post-training defense

| Dataset | Hit@10 (TR) | Hit@10 (IR) |
|---|---|---|
| Flickr-PASCAL | 0.978 (-0.006) | 0.954 (-0.017) |
| COCO | 0.976 (+0.065) | 0.945 (+0.109) |

94,313 images and 4,100,413 snippets of text (43.5 per image), each grounded to a region of an image. Figure 5b shows the results of post-training defense on the Attack II poisoned models on both datasets. We observe that the defense already shows effectiveness even with only one epoch. For example, on Flickr-PASCAL, the Hit@10 drops from 0.9 to around 0.0 at the first epoch, and remains at a very low level afterward. This shows the effectiveness of our defense. Note that the models' utility does not drop after the defense as shown in Table 8.

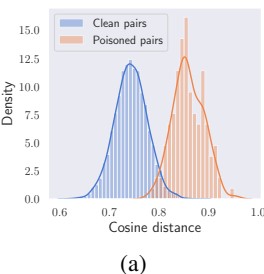

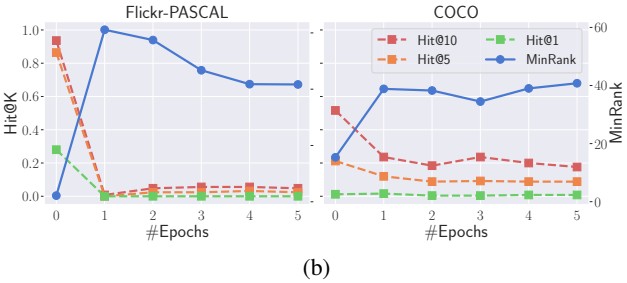

(a)             (b)

Figure 5: (a) Probability density of cosine distances between clean/poisoned pairs in Flickr-PASCAL. (b) Performance of post-training defense against Attack II poisoned models.

## 6 CONCLUSION

In this paper, we first study the vulnerability of data poisoning attacks against multimodal models in both visual and linguistic modalities. Our three types of poisoning attacks show their effectiveness in achieving remarkable attack performance while keeping the model's utility on clean data. Our evaluation of the poisoning effects on the visual and linguistic modalities shows that both modalities are vulnerable to poisoning attacks but reflected in different ways. Poisoning the visual modality leads to a better MinRank, while poisoning the linguistic modality results in higher Hit@K with a small K (e.g., 1). To mitigate the attacks, we propose two types of defenses. Our evaluation shows that both defenses effectively mitigate the attacks while preserving the multimodal model utility. To the best of our knowledge, our defenses are the first to address the data poisoning attack against multimodal encoders. In the future, we plan to extend our work into more different modalities.

REPRODUCIBILITY STATEMENT

To be reproducible for the findings of a study means that results and the observations should be achieved again with a high degree of reliability. Since our attacks only modify the training data and the training process is open-source, it is easy to conduct the experiments and reproduce our results. And we have comprehensively evaluated the attacks as well as the defenses across various dimensions, and it would be enough to follow our setting and conduct new methods based on them. To make it more reliable, we fix the random seed to 42 in our experiments. Regarding the required computational resources, since we only fine-tune the pretrained model instead of training it from scratch, it is relatively easier to follow the same settings as ours.

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

## A APPENDIX

### A.1 DATASET

In the experiments, we utilize 4 image-caption datasets to evaluate our techniques, including Flickr30k (Young et al., 2014) (abbreviated as Flickr), PASCAL (Rashtchian et al., 2010), COCO (Chen et al., 2015), and Visual Genome (VG) (Krishna et al., 2017). Flickr, PASCAL, COCO, and VG are four widely used benchmark datasets for various natural language processing and computer vision tasks. To explore the effect of the size of the dataset, we randomly select 50% (25%) samples from each class of COCO's training data to form the COCO-M (COCO-S) dataset. We keep the same test dataset for them, i.e., all sharing the same 3,900 images. Note that we combine Flickr and PASCAL as the training dataset Flickr-PASCAL, since Flickr contains no label information but has a large number of pairs and PASCAL has only a limited amount of labeled pairs.

*Flickr-PASCAL.* The Flickr dataset (Young et al., 2014) is a large-scale benchmark collection for sentence-based image description and search. It contains captioned images scraped from Yahoo's photo album website, Flickr, but has no class labels. The PASCAL dataset (Rashtchian et al., 2010) is a standard caption evaluation dataset containing 1,000 images with 20 categories. The PASCAL dataset is a balanced dataset, i.e., each class is represented with 50 images and each image is paired with 5 text captions. We divide the PASCAL dataset evenly into two parts, training and testing, at a rate of 1:1, thus keeping the balance at the same time. Since the PASCAL dataset is too small, we combine the training data of PASCAL and Flickr together as Flickr-PASCAL to train the model.

*COCO.* The COCO dataset (Chen et al., 2015) is one of the most representative large-scale object detection, segmentation, and captioning datasets. It has 80 object categories and contains 5 captions per image. For each image, we randomly select one of the object categories as its label; the more objects it contains, the more possible the object will be chosen. And we sampled and examined the label of the images and found them reasonable (Appendix). We count the number of images in each class in the COCO dataset. To make the dataset more balance, we remove the two classes with the lowest number, `toaster` and `hair drier`, which have 28 and 53 images, respectively. For the test data, we randomly choose 50 images with their captions from each class, and the test data contains 3,900 images with 78 classes.

*COCO-M/COCO-S.* The COCO-M/COCO-S dataset is a subset of the COCO dataset. We randomly select 50% (25%) samples from each class of COCO's training data to form the COCO-M (COCO-S) dataset. For the test data, we use the same test data as the COCO dataset, which contains 3,900 images with 78 classes.

*Visual Genome.* The Visual Genome (VG) (Krishna et al., 2017) dataset is a widely used region captions dataset. It contains 94,313 images and 4,100,413 snippets of text (43.5 per image), each grounded to a region of an image. We randomly select at most 5 texts for each image and form the training data. Note that we only use this dataset for fine-tuning in the post-training defense.

### A.2 MODEL STATISTICS

The statistics of our used CLIP model can be found in Table 9. CLIP-ViT-L/14 is the largest model. And CLIP-ViT-B/16 is larger than CLIP-ViT-B/32 in FLOPs while is slightly smaller than that regarding the number of parameters.

Table 9: Model size

| Model | FLOPs | # Params |
|---|---|---|
| CLIP-ViT-B/32 | 4.885G | 84.225M |
| CLIP-ViT-B/16 | 13.208G | 82.456M |
| CLIP-ViT-L/14 | 56.255G | 258.721M |

A.3   CASE STUDY FOR THE POOR PERFORMANCE OF SOME GOALS ON FLICKR-PASCAL.

As shown in Figure 6, each of image does not belong to person class. However, they all contain humans as their subjects. Their corresponding captions can even ignore their class. For example, in Figure 6, (a) is paired with "Two girls in pink and blue outfits." and "Two women pose beneath a sign saying Welcome to English Camp.", (b) is paired with sentences like "A family poses for a picture while out at a restaurant.", (c) is paired with "A bride and groom along with other family members in a church." and (d) is paired with "Three dark-haired young men sit in a classroom with one looking at his laptop.". These kinds of images can be easily found in the dataset, i.e., many images containing human subjects belong to other classes. Thus the person images are more than those labeled as person in the dataset, which implicitly lowers the poisoning rate and leads to lower attack performance.

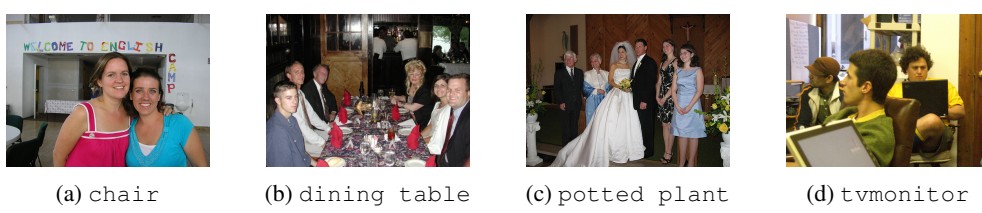

(a) chair          (b) dining table          (c) potted plant          (d) tvmonitor

Figure 6: Each image does not belong to the person category, but they all have human subjects.

