# OpenReview forum: "Data Poisoning Attacks Against Multimodal Encoders"
_ICLR.cc/2023/Conference — Submitted to ICLR 2023_

### Official Review · Reviewer_ygKY · 2022-10-24

**Confidence:** 4
**Correctness:** 3
**Technical Novelty And Significance:** 2
**Empirical Novelty And Significance:** 3
**Recommendation:** 5

**Clarity, Quality, Novelty And Reproducibility:**

This work is easy to follow and explains its reasoning in a clear manner. The focus is interesting and the experiments are very thorough.

**Strength And Weaknesses:**

Strength:
1. To my knowledge, this is the first paper that tries poisoning attacks on a multimodal model during the fine-tuning phase. The scenario is realistic and easier to expand further for future experiments compared to training CLIP from the scratch. Using Image-Text retrieval is also a novel evaluation task for analyzing multi-modal attacks with more evaluation metrics like hit@k and minrank.
2. This paper isolates the linguistic and visual modalities and analyzes the attack's effect on them separately, providing an interesting insight that may lead to effective attacks with more focusing goals (lower minrank or higher hit@k rate).
3. The paper gives simple but effective defenses against the proposed attacks.
4. The paper considers a more general attack scheme compared to the previous works.

Weakness:
1. I do not see significant differences between attack 2 and attack 3. both datasets seem to have consistent Hit@K ratios and MinRank values. The only values inconsistent are "ours-2" under the Flickr-PASCAL dataset, but I think it can be attributed to the specific dataset selections rather than the attack 3 itself being more harmful. There is an analysis in the ablation study on different combinations of classes during the attacks, but the consistency issue is not analyzed fully during the main experiment part.

2. I also cast a doubt on the dataset used, specifically Flickr-PASCAL. Even though the combined dataset has a relatively large size, only 20 categories from the PASCAL datasets could be selected to be poisoned and that may influence the scope of the attack. Namely, even though the overall poisoning rate is low, the effect may be stronger due to the data imbalance issue.

3. Combing the freezing experiment with the main experimental result, I can have a clear understanding of the linguistic modality being attacked. However, the analysis of which modality is most vulnerable is not satisfactory. From the experiment, each image is matched to more than one caption, rendering an imbalance in encoding representations. As stated in previous work, when there are more diverse caption sets for each image, the model is more likely to change the image encoder compared to the text encoder. Although the statement is unjustified theoretically, It makes sense intuitively and a more considerate experiment concerning this issue or an explanation of why this concern is unnecessary could be very helpful to the overall argument. There is also a lack of explanation and extension on their result with the difference in attacking different modalities.

4. Both the pre-training defense (removing less correlated data pairs) and the post-training defense sound a bit weak. For the pre-training, it is possible to have harder training tasks with larger cosine distances, and setting a threshold is difficult to do and could be harmful to the performances. For the post-training, the paper does not explore the situation when the fine-tuning datasets are of different scopes.

5. Overall, I think there is a lack of analysis and expansion on the experiments discussed

**Summary Of The Paper:**

This paper explores the poisoning attacks on CLIP. Compared to the previous work that focused on pre-training attacks on the image-modal encoder, this paper focuses on attacking the fine-tuning process and evaluating it using text-image retrieval, a downstream task that shows the attack's effect on both image and text-modal encoder. The attack is also generalized to a class-class poison rather than a target-class poison. The paper analyzes the differences between poisoning effects on both modalities and proposes post- and pre-training defenses against these attacks.

**Summary Of The Review:**

The paper gives many new insights into analyzing attacks on multi-modal models. The training scheme on fine-tuning tasks makes future experiments more manageable and easier to transfer. However, I think there is a lack of explanation for many of the findings. I will raise my rating if there are more explanations on the points I mentioned.

---

> ### Author Response · Authors · 2022-11-11
> **Author Response**
>
> We thank the reviewer for their questions:
>
> 1. Difference between Attack 2 and 3.
>
> We agree that attack 2 and attack 3 are similar, the difference is that in attack 2 we only inject one poisoned pair (e.g., sheep2aeroplane). However, in attack 3, we inject multiple poisoned pairs (e.g., sheep2aeroplane and boat2dog). In this case, attack 3 is used to demonstrate that the model is able to “remember” multiple poisoned pairs with a one-time injection of poisoned samples.
>
> 2. The Flickr-PASCAL dataset.
>
> Thanks for your question, we also considered the issue of data imbalance. We analyzed the two datasets and found that they have similar scopes. For example, in PASCAL we have the label dog. In Flickr, there are also many text-image pairs that are related to dogs. The main difference is that the Flickr dataset is larger and has no labels. Thus there are a large number of samples in Flickr belonging to the same class of PASCAL, which means the data imbalance should not be an issue in this case.
>
> 3. The imbalance issue of datasets.
>
> Thanks for your advice.
> We conduct an experiment on both datasets. We randomly select one caption for each image to prevent the imbalance issue and make the comparison more reliable. Besides, we keep the other settings the same as the experiments in the paper. The table shows that poisoning attacks achieve good performance on the balanced dataset.
>
> |Dataset|Hit@1|Hit@5|Hit@10|MinRank|
> |-|-|-|-|-|
> |Flickr-PASCAL|0.160|0.848|0.944|2.904|
> |COCO|0.048|0.392|0.712|11.372|
>
> Then we compare the difference of the clean and poisoned encoders by computing the cosine distance between the embeddings of the clean and poisoned encoders. The table below shows the differences when poisoning text and image encoders.
>
> |Dataset|Text|Image|
> |-|-|-|
> |Flickr-PASCAL|0.044|0.047|
> |COCO|0.047|0.064|
>
> The results are comparable with Figure 1(b) in the paper, which shows that both encoders are influenced by the poisoning attack. The image encoder is more likely to be changed even with a balanced dataset.
>
> 4. Pre-training defense.
>
> We agree that on harder training tasks, the cosine distances could be larger. However, we emphasize that the cosine distance is relative and the threshold is dynamically selected by the defender. That is, on harder training tasks, the cosine distances between mismatched pairs could be even larger (otherwise the model performs really poorly on this task).  We can further dynamically select the threshold and only remove up to a certain portion of samples. For example, if we remove at most 10% of potentially poisoned data, the original task is supposed to maintain similar performance.
>
> 5. Post-training defense.
>
> First, the multimodal model we used, i.e., CLIP, is trained on a huge dataset that covers various scopes of images and their captions, which is relatively comprehensive. Second, such pre-trained models usually provide the pre-trained datasets or mention what kind of data is used to train the models. In that sense, the defender can use the same pre-trained dataset or collect a similar distribution dataset from the Internet to fine-tune the model as well.

---

### Official Review · Reviewer_zCXF · 2022-10-25

**Confidence:** 4
**Correctness:** 3
**Technical Novelty And Significance:** 1
**Empirical Novelty And Significance:** 3
**Recommendation:** 6

**Clarity, Quality, Novelty And Reproducibility:**

This paper is well-written and easy to follow. As for its novelty, it is new to see work on the poisoning attack domain to conduct a study on the linguistic modality. As for reproducibility, although the authors provide a reproducibility statement, it is still better to provide codes for other readers to implement the experiments.

**Strength And Weaknesses:**

Strength:
1. This paper investigates a new problem about whether attacking the linguistic modality is also effective for the poisoning attack task.
2. The experiments are extensive and provide a deeper understanding of the vulnerability of multimodal encoders.

Weaknesses:
1. There is a lack of baselines in this paper. Although the paper proposes three types of attack, the comparison does not include other state-of-the-art methods. As mentioned by the author, there are existing methods in this domain, but they are not introduced in the experiment to show whether the attacking method is not redundant.
2. There is a lack of discussion on whether it’s worth of attacking linguistic modality. Although it can be as effective as attacking linguistic modality, it is not clear whether it changes a lot of linguistic data compared to visual data. If not, based on the principle of not causing much difference in the data, it does not make sense to change the linguistic data.

**Summary Of The Paper:**

This paper focuses on the data poisoning task on multimodal encoders. This paper investigates three types of poisoning attacks first. After that, it studies the effectiveness of attacking visual and linguistic features. In addition, it explores two types of defense mechanisms for defending against the attack on multimodal encoders.

**Summary Of The Review:**

In total, this paper is novel for providing a study on attacking linguistic modality for multimodal encoders. The experiment is comprehensive and provides an easy-to-follow structure for readers. However, this paper lacks an illustration of why linguistic modality matters for poisoning attacking and whether the used attack method is better than the baselines. Thus, the reasonableness for conducting this type of attack needs more explanation.

---

> ### Author Response · Authors · 2022-11-11
> **Author Response**
>
> We thank the reviewer for the helpful suggestions:
>
> 1. Lack of baselines.
>
> As mentioned in the paper, we are the first to study the poisoning attacks against multimodal models with both image and text encoders. Thus, there is no suitable baseline for this problem. Our goal is to explore whether the linguistic and visual modalities are vulnerable under such a multimodal setting. Extensive experimental results show that our attack performs well with a relatively low poisoning rate. In this sense, our method could be considered as a baseline for future work which aims at achieving a higher poisoning performance with more advanced methods.
>
> 2. Whether it is worth attacking linguistic modality.
>
> Thanks for your suggestion.
> We believe it is worthwhile, and even crucial, for us to explore attacking linguistic modality in such a multimodal model. With the development of such multimodal models, there are many applications based on them, for example, text-to-image search engines or text2image generation models. Imagine, a user who wants to search for or generate images given the text “a lovely kid playing with a dog”, with a poisoned multimodal model as a backbone, may get many hateful images that contain violence, sex, or racial discrimination. Also, for applications like image captioning, if the linguistic modality is poisoned, the generated caption for an image could contain hate words or maliciously targeted sentences that could damage society. Therefore, it is important to study the vulnerability of linguistic modality to poisoning attacks because it is as important as the visual modality in our daily life.
>
> 3. Whether it changes more linguistic data than visual data.
>
> To construct the poisoned dataset, we mismatch the target images with target texts. In this way, both visual and linguistic data are changed by the same poisoned rate. Concurrently, we set the poisoning rate as 0.08% for Flickr-PASCAL and 0.24% for COCO, which indicates that we obey the principle of not causing much difference in the data.
> After training on the poisoned dataset, both image and text encoders are poisoned. To study which modality changes more, we compute the cosine distance between text and image embeddings of the poisoned and clean encoders, respectively. As illustrated in Figure 1(b) in the paper, the text embeddings of clean and poisoned models are more similar than the image embeddings on both datasets. In other words, the image embeddings change slightly more after poisoning.
>
> 4. Reproducibility.
>
> Regarding the code, we will make it publicly available after the acceptance of the paper.

---

### Official Review · Reviewer_KeYj · 2022-10-25

**Confidence:** 4
**Correctness:** 3
**Technical Novelty And Significance:** 3
**Empirical Novelty And Significance:** 3
**Recommendation:** 6

**Clarity, Quality, Novelty And Reproducibility:**

To my knowledge, this is one of the first work to explore attack/defense from both modalities in joint vision-language model. However, the attack and defense methods are simple and not entirely novel. Previous works have also shown vulnerability of CLIP and text model in data poisoning attacks, which also reduce the novelty and potential impact of this work.

**Strength And Weaknesses:**

Strength:
+ The paper is one of the first works to explore data security in both modalities of a joint vision-language model
+ Both the attacks and defense methods are simple but effective, with strong experimental results.

Weakness:
+ There has been numerous works on adversarial attacks on text models, many of them also about data poisoning, so it has been quite clear that linguistic modality is also vulnerable to data poison attack [1,2]
+ As Carlini et al. and others have shown in their previous work, CLIP is quite vulnerable to adversarial attacks, and they have also shown that CLIP can be poisoned with a very limited number of poisoned data points.
+ The attacks and defense method proposed in the paper is different compared to previous work of Carlini, but the novelty is limited.

[1] https://aclanthology.org/2021.naacl-main.13.pdf
[2] https://aclanthology.org/2021.naacl-main.13.pdf

**Summary Of The Paper:**

The paper examines data poisoning attacks and defense for joint language-vision model (CLIP) in a retrieval setting. Expanding on Carlini et al. 2022, the authors propose attacks and defenses on both modalities instead of just vision signal and show vulnerability in both.

The paper proposes three attacks, all try to lead the model to mis-associate the text signal to the image signal. The first attack targets a class of text, swapping out the image from an <image, text> pair for another image of a different class. The second attack generalizes the target of the first into a class of image, and the third attack generalize the second attack into multiple classes of images. All three attacks work well during the fine-tuning process on COCO and Flickr-PASCAL even with very low poisoning rate and with virtually no loss in term of original test data utility.

The authors then analyze the difference in the effect of data poisoning on the text/image encoders by freezing each and calculate the performance (Hit and Minimum rank) and show that the image and text encoder lead to slightly different forms of poisoned behavior. While the poisoned image encoder generally leads to worse behavior overall (generally higher rank of the poisoned image in retrieval), the poisoned text encoder leads to higher probability of top retrieval results being the poisoned class.

In the defense, the author proposes two methods, the first is data filtering with distance from the original encoders and the second is retraining with clean dataset. Both method improves the robustness substantially.

**Summary Of The Review:**

My main concern is about the potential impact of the paper. The previous work of Carlini has shown the vulnerability in CLIP model while the attack and defense method is not novel enough.

---

> ### Author Response · Authors · 2022-11-08
> **Author Response**
>
> We thank the reviewer for raising these points:
>
> 1. Existing works on text models.
>
> Previous works focus on the text model that is trained with only textual data. However, our target model is the multimodal model, which is jointly trained with textual and image data in a self-supervised way. Thus, we aim at demonstrating that both visual and linguistic modalities in the multimodal model are vulnerable to poisoning attacks. And this differs from the previous attacks against text models.
>
> 2. Differences with Carlini et al. and others.
>
> In previous works, they show that CLIP is vulnerable to poisoning attacks with a very limited number of poisoned data samples. However, they only focus on the downstream image classification task, i.e., they extract the image encoder and aim at misleading the downstream classifier built on the encoder. Our work is based on both text and image encoders, and the nature of our poisoning goal is to mismatch the embeddings of some targeted images to some targeted texts. In this sense, our work is the first to explore the vulnerability of linguistic modality in a multimodal model and the first to study the text-image alignment of the multimodal model.
>
> 3. Novelty.
>
> Considering the novelty, our goal is to explore whether the linguistic and visual modalities are vulnerable to poisoning attacks. In this sense, we are the first to study the security risk of multimodal models from the view of both visual and linguistic modalities.  More importantly, we are the first to propose defense methods against the poisoning attack in multimodal encoders, which Carlini et al. and others have not explored. As previous works and ours have shown the vulnerability of multimodal models to these attacks, we believe it is urgent to develop effective defense mechanisms against such attacks.

---

> > ### Comment · Reviewer_KeYj · 2022-11-22
> > **Thanks for the clarifications**
> >
> > I would like to thank the authors for the additional information on the paper. In my opinion, it would help in clarity if the authors state at the beginning that the paper focuses on just the alignment between the visual and text encoders, and not on the encoders themselves on downstream tasks.
> > I also agree with reviewer ygKY that the defense is quite weak. However, as one of the first works in security risks for this class of models, I still think the paper would have a reasonable contribution to future research.

---

### Decision · Program_Chairs · 2023-01-20

**Decision:**

Reject

**Justification For Why Not Higher Score:**

Explained in part 1.

**Justification For Why Not Lower Score:**

N/A

**Metareview: Summary, Strengths And Weaknesses:**

The paper studies vulnerability of language and image modalities in multimodal language-vision pretrained models (CLIP) against poisoning attacks, during fine-tuning. In particular, the paper conducts three types of poisoning attacks and then proposes pre- and post-training defenses. The paper was discussed in the meeting with the reviewers, which I summarize below.

Main strength: All the reviewers acknowledged the importance of the problem and the necessity of studying and defending attacks on multimodal models.

However,
- The main concern is that the proposed attacks are applied to and tested on the same data. This makes the proposed attacks (based on e.g. changing the labels) trivial to detect. A more interesting scenario is when the attack success rate is measured on (more than two) different datasets.
- Proposed defenses are very simple and may not be always effective.
- The quality of every encoder should be studied alone, to make sure that the attack does not affect the quality of each encoder on the rest of the data.
- Attacks during training from scratch is more realistic, and fine-tuning data is usually clean and more difficult to attack.
The above points will make the paper much stronger and more convincing.